# Measurable signatures of bosonic fractional Chern insulator states and their fractional excitations in a quantum-gas microscope

**Botao Wang[1]⋆, Xiao-Yu Dong[2]† and André Eckardt[1]‡**

**1** Institut für Theoretische Physik, Technische Universität Berlin,
Hardenbergstraße 36, 10623 Berlin, Germany
**2** Department of Physics and Astronomy, Ghent University,
Krijgslaan 281, 9000 Gent, Belgium

⋆ botao.wang@tu-berlin.de, † xiaoyu.dong@ugent.be, ‡ eckardt@tu-berlin.de

## Abstract

The recent progress in engineering topological band structures in optical-lattice systems makes it promising to study fractional Chern insulator states in these systems. Here we consider a realistic finite system of a few repulsively interacting bosons on a square lattice with magnetic flux and sharp edges, as it can be realized in quantum-gas microscopes. We investigate under which conditions a fractional Chern insulator state corresponding to the Laughlin-like state at filling $\nu = 1/2$ can be stabilized and its fractional excitations probed. Using numerical simulations, we find an incompressible bulk density at the expected filling for systems, whose linear extent is as small as 6-8 sites. This is a promising result, since such small systems are favorable with respect to the required adiabatic state preparation. Moreover, we also see very clear signatures of excitations with fractional charge in response both to static pinning potentials and dynamical flux insertion. Since the compressible edges, which are found to feature chiral currents, can serve as a reservoir, these observations are robust against changes in the total particle number. Our results suggest that signatures of both a fractional Chern insulator state and its fractional excitations can be found under realistic experimental conditions.



# 1  Introduction

In recent years, we have seen tremendous progress in realizing artificial gauge fields and topologically non-trivial band structures in systems of ultracold atoms [1–5]. This includes, the realization of artificial magnetic fields [6–9] and spin-orbit coupling [10, 11], the measurement of the quantum anomalous Hall effect [12] and topological invariants [13–20], as well as the observation of chiral edge currents [21–29]. Since atomic quantum gases in optical lattices are highly controllable, especially allowing for the ability to manipulate and observe the system with single-lattice-site resolution in the so-called quantum-gas microscopes [30–33], it is an intriguing perspective to combine the realization of topologically non-trivial band structures with strong interactions in these systems, for the purpose of stabilizing fractional Chern insulator (FCI) states (i.e. the lattice analogues of fractional quantum Hall states) [34, 35], as well as to probe and manipulate individually their anyonic excitations.

As a paradigmatic model that describes a square lattice with homogeneous magnetic fluxes, the Harper-Hofstadter (HH) model is known to exhibit topologically non-trivial bands [3, 36, 37]. In the presence of strong interactions, up to the limit of hard-core bosons, this system is predicted to host FCI ground states [38–44]. Various realistic protocols were proposed for both the detections of the FCI states [45–57], and their adiabatic preparations [57–68] (see also Ref. [69] for rapid state preparation via magnetic flux ramps). Experimentally, the HH model has been realized by using bosonic atoms in optical lattices [13, 70, 71]. Moreover, the dynamics of two interacting particles has recently been investigated in a ladder geometry [25] without encountering driving induced heating (see, e.g., [72]) on the experimental time scale. These achievements suggest that it will be possible to stabilize and probe fractional Chern insulator states of (at least a few) interacting bosons on a lattice in a quantum-gas microscope.

In this paper, we consider realistic system geometries with open boundary conditions with a quarter of a flux quantum per plaquette. We focus on the regime, where a Laughlin-like state at a filling of $\nu = 1/2$ particles per flux quantum is expected. Using numerical simulations based on DMRG [73–77], we compute experimentally accessible quantities and explore in which parameter regimes they show signatures of a FCI state. In particular, we (i) investigate the minimal system sizes required to show homogeneous bulk behavior at the filling factor expected for the FCI state, (ii) find that the edges serve as a reservoir for particles allowing for variations of the total particle number, (iii) show that both static local pinning potentials and dynamical flux insertion can be employed to probe charge fractionalization, and (iv) point out that FCI behavior is robust against various parameter variations. Our results are complementary to those of the recent paper by Repellin et al. [50], which discusses FCI signatures in the Hall drift and the density response to variations of the magnetic flux.

## 2 Model

We consider strongly-interacting/hard-core bosons in a two-dimensional (2D) square lattice, described by the Harper-Hofstadter-Hubbard model under Landau gauge:

$$\hat{H} = \sum_{m,n} \left( -J_x e^{-i\phi n} \hat{a}^{\dagger}_{m+1,n} \hat{a}_{m,n} - J_y \hat{a}^{\dagger}_{m,n+1} \hat{a}_{m,n} + \text{h.c.} \right)$$

$$+ \frac{U}{2} \sum_{m,n} \hat{n}_{m,n}(\hat{n}_{m,n} - 1) + \sum_{m,n} W_{m,n} \hat{n}_{m,n}. \tag{1}$$

Here $\hat{a}^{\dagger}_{m,n}(\hat{a}_{m,n})$ are the bosonic creation (annihilation) operators and $\hat{n}_{m,n} = \hat{a}^{\dagger}_{m,n}\hat{a}_{m,n}$ are the number operators on sites $(m,n)$, with integer site indices $m = 0, 1, \ldots, L_x - 1$ and $n = 0, 1, \ldots, L_y - 1$ along the directions $x$ and $y$, respectively. They define a rectangular system of $L_x \times L_y$ sites with open boundary conditions that will be occupied by $N$ particles. The hopping terms between nearest neighbors has the strength $J_x$ ($J_y$) in the $x$ ($y$) direction. The Peierls phase factors for the hoppings in $x$-direction give rise to a magnetic flux of $\phi = 2\pi\alpha$ per plaquette. In this work, we mainly focus on $\alpha = 1/4$ since the Harper-Hofstadter model with this value has been readily realized in cold atom experiments [13, 21, 25, 71]. Here h.c. refers to the Hermtian conjugations. The Hubbard parameter $U$ quantifies the on-site interactions between the particles. In the following we assume hard-core bosons, where an infinitely large positive value of $U$ suppresses doubly occupied lattice sites completely (We have also checked that qualitatively similar results are found for large values of $U$, as exemplified in Section 5). Finally, we also consider different on-site potentials $W_{m,n}$ that will be specified later. The energies and angular frequencies will be measured in unit of $J_x$, i.e. $\hbar = 1$ and $J_x = 1$.

## 3 Ground state properties

The Hofstadter model for single particles is a paradigmatic model that exhibits topological non-trivial bands characterized by non-zero Chern numbers [3]. By introducing strong interactions and partially filling the lowest band, the ground state is predicted to be a FCI state analogous to a Laughlin state [38–41, 43, 45–50, 65, 66]. In this section, we explore signatures of $\nu = 1/2$ FCI state which could be the ground state with flux $\alpha = 1/4$ in open boundary conditions and $W_{m,n} = 0$. For this purpose, we employ the DMRG algorithm from the TenPy library [77].

As a first experimentally observable quantity, we study the ground-state density $n_{m,n} = \langle \hat{n}_{m,n} \rangle$. We choose the particle number $N$ which satisfies the half filling condition $N/N_\phi = \nu = 1/2$, where $N_\phi = \alpha(L_x - 1)(L_y - 1)$ is the number of flux quanta piercing the lattice. Note that for a finite system with open boundary conditions $N_\phi$ is noticeably different from the value $N'_\phi = \alpha L_x L_y$ found for periodic boundary conditions. The corresponding incompressible FCI ground state is expected to feature a uniform density distribution in the bulk, with an average density of $\rho = \alpha\nu = 1/8$ particles per site. In Fig. 1(a) we plot the density distribution of a system with $N = N_\phi/2 = 14$ particles on $17 \times 8$ sites. We can observe that this finite system features already a flat density distribution at the expected bulk density $\rho = 1/8$ in its center. This can be seen more clearly also in the upper panel of Fig. 1(b), where we plot the density $n_{m,3}$ along the central row ($n = 3$) versus the $x$-coordinate $m$. This is a first indication of the expected incompressible behavior of the FCI state. Further evidence for incompressibility is gathered when plotting the density also for different total particle numbers in Fig. 1(b): While the density increases close to the boundary, the bulk remains at $\rho = 1/8$ for up to $N = 16$ particles. Thus, compressible boundaries serve as a reservoir for the center of the system. For $N \geq 17$ or $N \leq 10$, we find noticeable density oscillations also in the center, suggesting a break-down of the FCI state. This observation also confirms that we should,

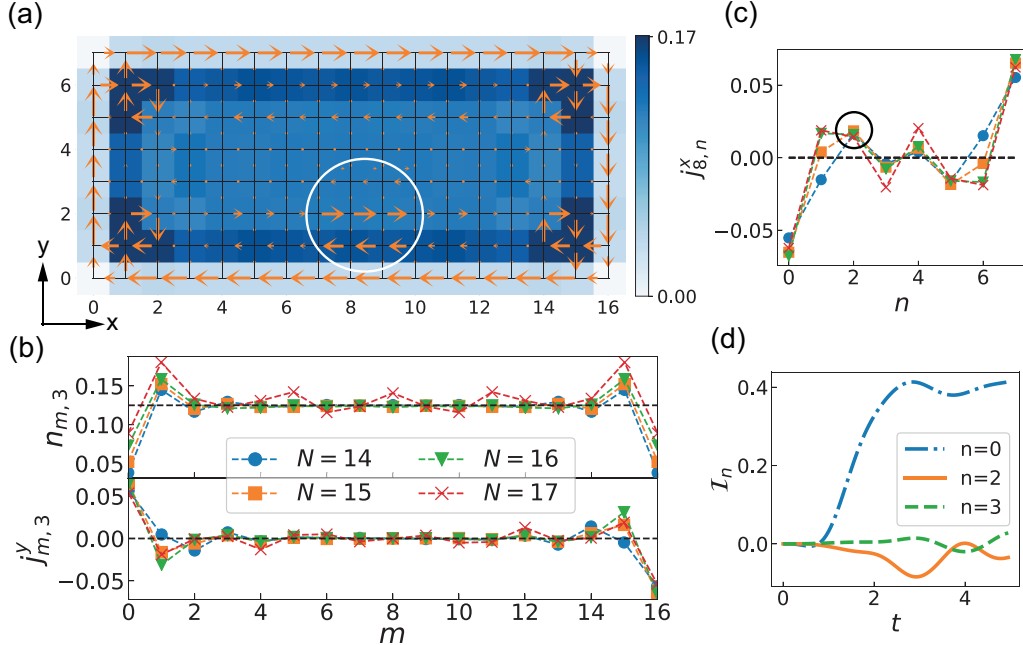

Figure 1: (a) Spatial density and current distributions of the ground state of hard-core bosons in a lattice of size $17 \times 8$ with $N = 14$, $\alpha = 1/4$ and $J_y = 1$. The magnitudes of current in the white circle are zoomed in by a factor of 3 for a clear visualization. (b) Densities (upper panel) of the middle row $n = 3$ and vertical currents (lower panel) on the middle links connected by $n = 3$ and $n = 4$. The horizontal dashed lines locates the expected $\rho = 1/8$ and $j = 0$. (c) Horizontal currents on the middle bonds connected by $m = 8$ and $m = 9$. The legend is the same as that in (b). The black circle is used to highlight that the currents with opposite chirality are almost independent of $N$. (d) Imbalance in the $n$-th row $\mathcal{I}_n$ as a function of evolution time.

indeed, consider particle numbers $N$ that are close to $N_\phi/2 = 14$, rather than ones close to $N'_\phi/2 = 17$.

A second observable that we study is the particle current between neighboring sites. The current leaving site $(m, n)$ in positive $x$ and $y$ direction is given by the operators

$$\hat{j}^x_{m,n} = iJ_x \left( e^{-i\phi n} \hat{a}^\dagger_{m,n} \hat{a}_{m+1,n} - \text{h.c.} \right), \tag{2}$$

$$\hat{j}^y_{m,n} = iJ_y \left( \hat{a}^\dagger_{m,n} \hat{a}_{m,n+1} - \text{h.c.} \right), \tag{3}$$

respectively, as can be confirmed by writing out the time derivative of $n_{m,n}$ [78]. We denote their mean by $j^\eta_{m,n} = \langle \hat{j}^\eta_{m,n} \rangle$ with $\eta = x, y$. Such currents have been measured recently in systems of ultracold atoms in optical lattices [21]. The $\nu = 1/2$ FCI state is expected to feature chiral edge currents in the presence of open boundary conditions. This behavior is also confirmed in Fig. 1(a), where the currents are indicated by orange arrows, whose direction and size indicate direction and magnitude of the current, respectively. In the lower panel of Fig. 1(b), we plot the $y$ current in the center ($n = 3$), $j^y_{m,3}$, as a function of the $x$-coordinate $m$. We can see that to a good approximation it vanishes in the center of the system, where we also found signatures of incompressibility in the density distribution.

An interesting observation is that when moving inwards from the boundary to the bulk, the chiral current not only decays in magnitude, but also reverses its sign in an oscillatory fashion. In order to make this effect more visible, we have enlarged the size of the current arrows by a factor of 3 inside the white ring in Fig. 1(a). It can also be seen in Fig. 1(c), where the typical

currents $j^x_{8,n}$ in center horizontal links (between the sites $m = 8$ and $m = 9$) are plotted versus the $y$-coordinate $n$. For various particle numbers, it is a robust observation that the current at the second row ($n = 2$) away from the boundary ($n = 0$) is opposite to the current at the boundary. Another feature, which seems to be related to these edge current oscillations, is the formation of current vortices around the plaquettes close to the corner. Similar behavior also shows up in larger systems (see Appendix A).

The current can be measured in various ways [21, 79–86], for example, by suddenly isolating two neighboring sites and subsequently observing the change of the density imbalance in linear order with respect to time. It can also be inferred from the spatial density imbalance $\mathcal{I}_n$ in the $n$-th row of the system [45],

$$\mathcal{I}_n(t) = \sum_{m < L_x/2} n_{m,n}(t) - \sum_{m > L_x/2} n_{m,n}(t), \tag{4}$$

as it builds up with time after releasing a single particle from the center of that row. Such an extra particle can be created by preparing the ground state of the system in the presence of a large potential dip ($V = -20$) at site ($m = 8, n$), which is then suddenly switched off. The time evolution of $\mathcal{I}_n$ for this scenario is plotted in Fig. 1(d). The change of the imbalance that builds up for short times ($t \lesssim 4$) directly after the quench is consistent with the computed ground state currents.

For the adiabatic preparation of FCI insulator states, it will be favorable to consider small systems [65,66,87]. In order to estimate the minimal linear extent permitting the observations of FCI signatures, we compute the ground state for systems of different widths $L_y$. Keeping $L_x = 17$ and $\alpha = 1/4$ fixed and targeting the $\nu = 1/2$ state, the particle number is always chosen as $N = \nu N_\phi = 2(L_y - 1)$. For small $L_y$ the system can be viewed as a flux ladder, as they were investigated recently in various experiments [20, 22–29]. In this one-dimensional (1D) limit the FCI states are predicted to be adiabatically connected to charge density waves (CDW) [62, 87–92]. The spatial density and probability current distributions are plotted in Fig. 2(a). One can observe density oscillations are consistent with CDW behavior in the "bulk" (i.e. the inner sites) for $L_y$ smaller than 5. When $L_y \gtrsim 6$, a homogeneous bulk density is formed at the filling of $\rho = 1/8$ particles per site expected for the FCI state. In order to quantify this statement we plot the mean [Fig. 2(b)] and standard deviation [Fig. 2(c)] of the density $n_{m,n}$, averaged over the central sites ($m = 4, \ldots 8$) of the middle row (e.g. $n = 2$ for $L_y = 5$). One can clearly see that the filling approaches $1/8$ for $L_y \gtrsim 6$ and the standard deviation becomes (very) small for $L_y \gtrsim 6$ (8). This suggests that a linear extent of 6 to 8 lattice sites should be sufficient to observe FCI bulk behaviour.

## 4 Fractional excitations via pinning potentials

In this section, we investigate signatures of charge fractionalization. In order to probe the fractional "charge" of $1/2$ and $-1/2$ of the anyonic quasiparticle (QP) and quasihole (QH) excitations of the $\nu = 1/2$ FCI, we compute the ground state of the system in the presence of local potential dips and bumps of strength $-V$ and $V$ (to be specified below) by setting the corresponding local potential terms $W_{m,n}$. These lower the energy of localized quasiparticle and quasihole excitations, respectively, so that above a threshold value of $V$, we expect an excited state with such excitations to become the new ground state of the system. Such pinned excitations have been used to estimate the spatial extent of QHs in various FCI models [47,93,94] and also for extracting their anyonic statistics [48,95–99]. For probing the fractional charge of the excitations, we will compare the ground-state density distribution of the system with and without such pinning potentials. As a signature of the formation of the FCI, we expect the

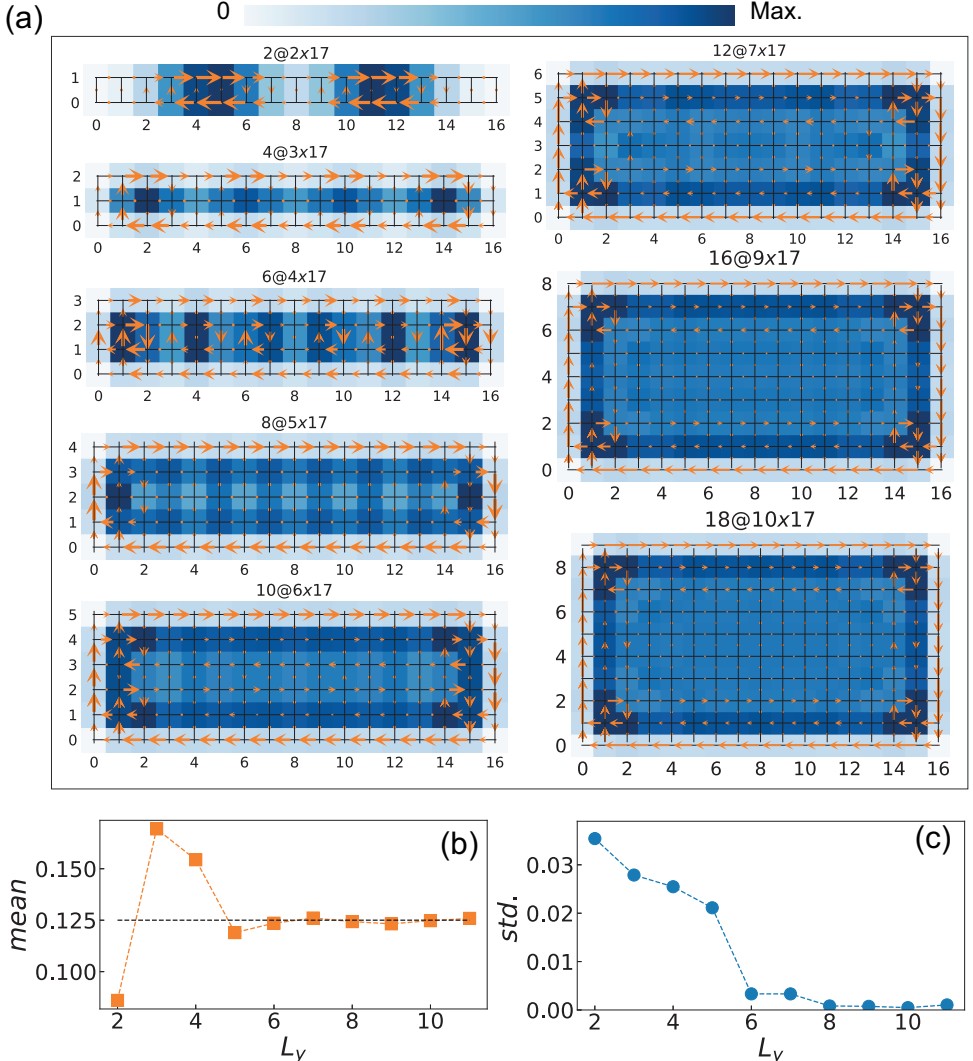

Figure 2: (a) Density and current patterns in the ground state for increasing values of the vertical extent $L_y$ show a crossover from CDW to FCI behavior. The mean (b) and the standard deviation (c) of the particle number density on the middle row with $m$ from 4 to 12. Other parameters are $\nu = 1/2$, $\alpha = 1/4$, $J_y = 1$, $L_x = 17$ and $N = 2(L_y - 1)$ in the hard-core limit.

particle number in the vicinity of these local defects to change in steps of the fractional charge $1/2$. The results presented here extend an earlier study based on exact diagonalization [46], which was limited to rather small systems of 4 particles only. We will see that not only the expected fractional charges can be observed very clearly, but also that this behaviour is robust against the variation of various system parameters, like the magnetic flux, the particle number, or a tunneling anisotropy. These results are very promising regarding a possible experimental observation of charge fractionalization in FCIs.

In the first set of simulations, we consider systems with $\alpha = 1/4$ and a horizontal extent of $L_x = 21$. We vary the vertical extent $L_y$ and the particle number $N$, so that $N$ takes integer values close to $\alpha\nu(L_x - 1)(L_y - 1) = (L_y - 1)5/2$. In the center of the left and right half of the system, respectively, we place a potential bump and a potential dip [as sketched in Figs. 3(c) and (f)]. Namely, on $2 \times 2$ or $3 \times 1$ neighboring sites, the potential is changed by $\pm V/4$ or $\pm V/3$ (In all later plots, we choose the $2 \times 2$ configuration). The choice of such a pair of defects is motivated by the desire to keep the average filling away from the defects constant. We will

see below, however, that the effect can also be observed for single defects, since the edge can provide/absorb the required charge. For every value of $V$, we then compute the ground state and compare how the particle number in the vicinity of the pinning potentials changes with respect to the ground state with a homogenous bulk ($V = 0$). We define the accumulated charge as

$$Q_{\pm V} = \sum_{\ell \in D_\pm(r)} \left[ \langle n_\ell \rangle_{\pm V} - \langle n_\ell \rangle_{V=0} \right]. \tag{5}$$

Here, the region $D_\pm(r)$ includes all the sites $\ell = (m, n)$ within a disc of radius $r$ that is co-centred with the local impurity. The radius $r$ has to be chosen big enough so that the regions $D_+(r)$ and $D_-(r)$ essentially contain the whole pinned QH and QP excitation, respectively. In sufficiently large systems the precise choice of $r$ should not matter, as long as this condition is fulfilled. This is indeed, what we observe [see Figs. 3(b) and (e) and the discussion below]. In the following we choose $r = 4$.

In Figs. 3(a) and (d) we plot $Q_\pm$ for various $L_y \geq 6$ as a function of $V$ for $2 \times 2$ and $3 \times 1$ pinning potentials, respectively. We can clearly observe two effects. Firstly, the density is hardly affected by a small pinning potential. This is another confirmation of the bulk incompressibility expected for the FCI state. Secondly, once $V$ is raised above a threshold, the accumulated charge quickly changes to values close to $\pm 1/2$ at which it stays to form an extended plateau with respect to $V$. This confirms the fractional charges of the elementary QP and QH excitations of the system. Moreover, the results not only show that it is a robust way to create QP and QH excitations by using pinning potentials, but also indicate that the

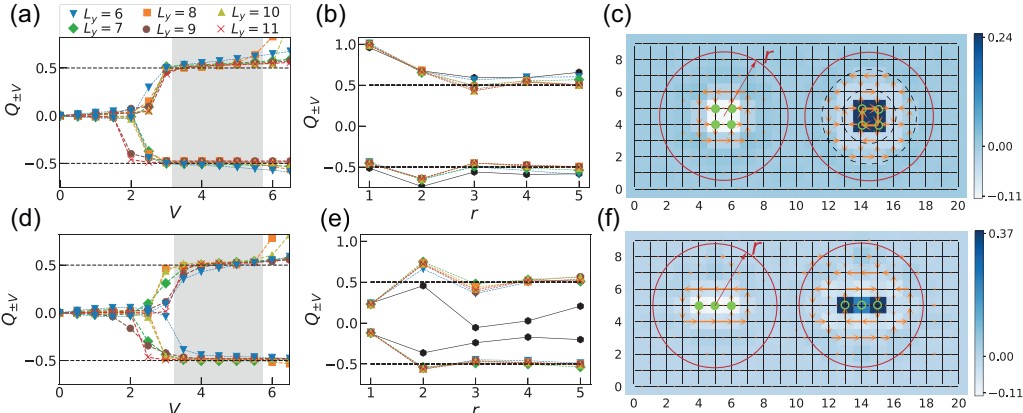

Figure 3: (a) Charges of QP/QH induced by negative/positive four-site pinning potentials as a function of pinning strength $V$. We have fixed $\nu = 1/2$, $L_x = 21$, $\alpha = 1/4$, $J_y = 1$ and adapted $N = (L_x - 1)(L_y - 1)\alpha\nu \approx 13, 15, 18, 20, 23, 25$ for $L_y = 6, 7, 8, 9, 10, 11$ respectively. (b) Integrated charges in the vicinity of negative/positive pinning potentials $|V| = 5$ as a function of the radius $r$ of the counted disc. The black hexagon dots correspond to $L_y = 5$, which shows a small system fails to give expected fractional charges. The legend in (b) is the same as that in (a) and lines are guide for the eye. (c) Distribution of the density and current differences between the ground states with $V = 5$ and $V = 0$ in a system of $21 \times 10$ with $N = 23$. The impurities of strength $\pm V/4$ are distributed over four sites of a plaquette, as indicated by the solid and empty green circles respectively. The red circles are used to locate the counting region $D(r)$ with radius $r$. The dashed circles with $r = 2$ and 3 capture the currents of opposite chirality. (d-f) Same plots as (a-c), but using a three-site pinning. The shaded area in (a) and (c) indicate the regime of pinning strength that is able to pin the expected fractional charges.

shape of topological excitations could be tailored by designing the pinning potentials. While we can identify plateaus already for $L_y = 6$, $Q_\pm$ remains closer to $\pm 1/2$ for larger system sizes. Choosing $V = 5$ as a value in the middle of the plateau, we compare results for different radii $r$ in Figs. 3(b) and (e) and find that they saturate close to $\pm 1/2$ for $r \geq 3$.

A typical density and current distribution, as it is found for $V = 5$ (in the middle of the plateau) is presented in Figs. 3(c) and (f) for $L_y = 10$. The extent of the QP is larger than that of the QH, which can be most clearly seen in the probability currents surrounding these localized excitations. When moving away from the center of the QP or QH, we observe that both the excess density and the chiral currents oscillate and change sign. These oscillations are more prominent for the QP excitation on the right hand side.

As discussed in the previous section, where we investigated ground-state properties without pinning potential, the edge of the system can serve as a reservoir for excess particles. Thus, we can expect that it is also possible to create not only charge-neutral pairs of QP and QH, but also individual excitations in the bulk. This scenario is investigated in Fig. 4. By applying only one potential dip in a system that otherwise agrees with the one studied in Fig. 3(c), we find the same signatures of incompressibility and charge fractionalization in the response of the ground-state density as before. The additional charge required for the creation of a QP by the potential dip is provided by the compressible edges. Note that this implies that the reservoir

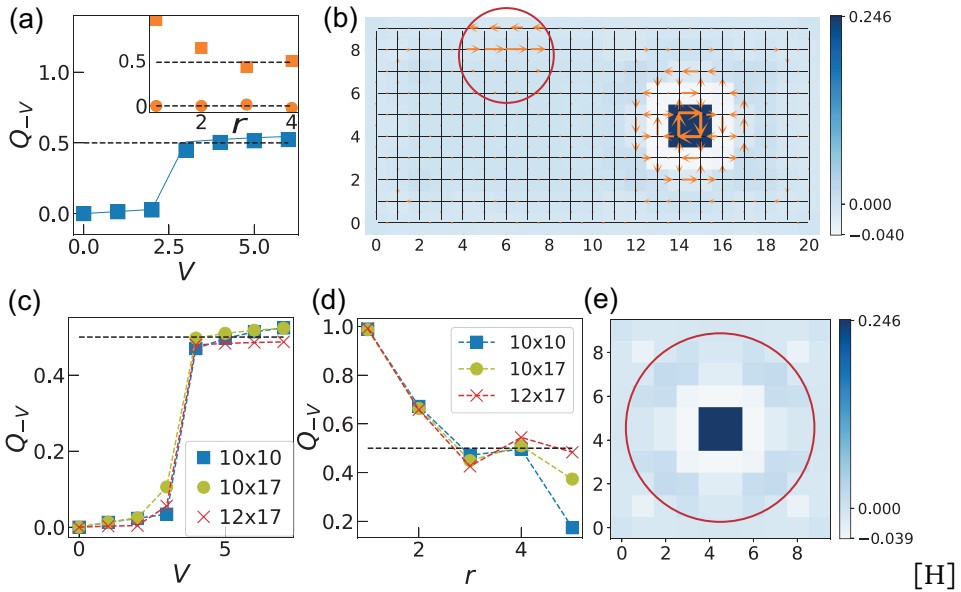

Figure 4: (a) Charges integrated in the vicinity of negative four-site pinning potentials at $r = 4$ as a function of $V$. The discrete points correspond to the case of a single potential dip, while the solid lines are obtained by applying both potential dips and bumps as in Fig. 3. The inset shows the charges at $V = 5$ as a function of the radius $r$, where the circles and squares respectively represent the charges in the left and right regions as defined in Fig. 3(a). We have used $N = 23$ in a system of $21 \times 10$. (b) A typical charge and current distributions at $V = 5$. The currents within the circle are zoomed in by a factor of 6 for clear visualization. (c) Charges as a function of $V$ for different systems with negative $2 \times 2$ pinning potentials. The numbers $N$ are given by 10, 18, 22 for system size $10 \times 10$, $17 \times 10$, $17 \times 12$, respectively, to achieve $\nu = 1/2$ particle per flux quantum. (d) Charges as a function of radius $r$ at $V = 5$. Note that the drop at $r = 5$ for $L_y = 10$ is due to extra excitations appearing in the edge. (e) A typical charge distribution at $V = 5$ in system of size $10 \times 10$ with $N = 10$. Other parameters are $\alpha = 1/4, J_y = 1$.

given by the compressible edges can also host (and thus exchange with the system) fractional charges in units of $v = 1/2$. Such a behavior can already be observed in a smaller system of $10 \times 10$ sites, see Fig. 4(c-e). By performing simulations with a single potential bump, we also find similar signatures for the creation of a single QH (not shown).

We will now investigate the robustness of the fractionalized ground-state response to pinning potentials, when changing various system parameters. Starting from a scenario like the one investigated in Fig. 3, with a pair of $2 \times 2$ pinning potentials of opposite sign and a system size of $21 \times 10$ sites, in Fig. 5 we show results for different total particle numbers, plaquette fluxes, and anisotropic tunneling matrix elements. In panel (a), we plot the accumulated charges $Q_{\pm V}$ as a function of $V$ for different particle numbers. Again the precise choice of $r = 4$ does not significantly influence the results, as can be inferred from (b), where $r$ is varied for fixed $V = 5$. Optimal particle numbers are expected to lie close to $v\alpha(L_x - 1)(L_y - 1) = 22.5$. And, indeed, we can see very clear signatures of charge fractionalization for a range of particle numbers $N = 21, 22, 23$. However, when the particle number becomes too small (large), the threshold value of $V$ at which a QH (QP) is created shifts to smaller values. Moreover, for the smallest particle number considered, $N = 20$, we even find the creation of a second QH excitation at a second threshold value of $V$.

The fact that the charge fractionalization expected for the FCI state breaks down when the global filling $N/N_\phi = N/[\alpha(L_x - 1)(L_y - 1)]$ deviates too much from the bulk value $v = 1/2$ of the FCI, can also be observed by varying the plaquette flux quantified by the number of flux quanta per plaquette $\alpha$. This is investigated in Figs. 5(c) and (d). In panel (c) we observe that the threshold for the creation of a QH (QP) is shifted to smaller values of $V$, when $\alpha$ increases (decreases). Moreover, for the value of $\alpha = 0.23$, no QP of fractionalized charge $1/2$ can be observed. In panel (d), we plot the pinned charges versus $\alpha$ (comparing different values of $V$) and find that the fractionalization of both QH and QP can be observed for $\alpha$ between 0.25 and 0.27. The quantization of QH (QP) alone can, moreover, be observed for values of $\alpha$ as small (large) as 0.24 (0.29).

Finally, we investigate the effect of a tunneling anisotropy. The robustness of charge fractionalization with respect to a variation of $J_y$ relative to $J_x = 1$ is investigated in Figs. 5(e) and (f) [1]. We find charge fractionalization for values of $J_y$ between 0.7 and 1.5. All in all, we can see that signatures of charge fractionalization can be observed in an extended parameter regime, which is good news for a possible experimental observation of charge fractionalization in small bosonic FCIs.

## 5 Effect of finite interactions

So far, we have focused on the hard-core limit. However, similar behaviour is found also for sufficiently strong, but finite interactions $U$. In order to test the robustness of charge fractionalization with respect to different interaction strengths, we have considered a system of $17 \times 8$ sites with 14 particles (like the one studied in Fig. 1) and calculated the response to two local defect potentials of oppositie sign [like the ones depicted in Fig. 3(c)]. For the calculation we truncated the maximum possible occupation of each lattice site to four particles. In Fig. 6, one can observe clear signatures of charge fractionalization for interaction strengths $U/J \gtrsim 15$.

---

[1] We consider values of $J_y$ that are both smaller and larger than 1. In systems with boundaries and defect positions that are symmetric with respect to both lattice directions, it would be sufficient to increase $J_y$ relative to $J_x = 1$. However, since we are working in a rectangular system that is elongated in $x$ direction and possesses two defects that are separated in $x$ direction, increasing and decreasing $J_y$ from 1 can lead to different results.

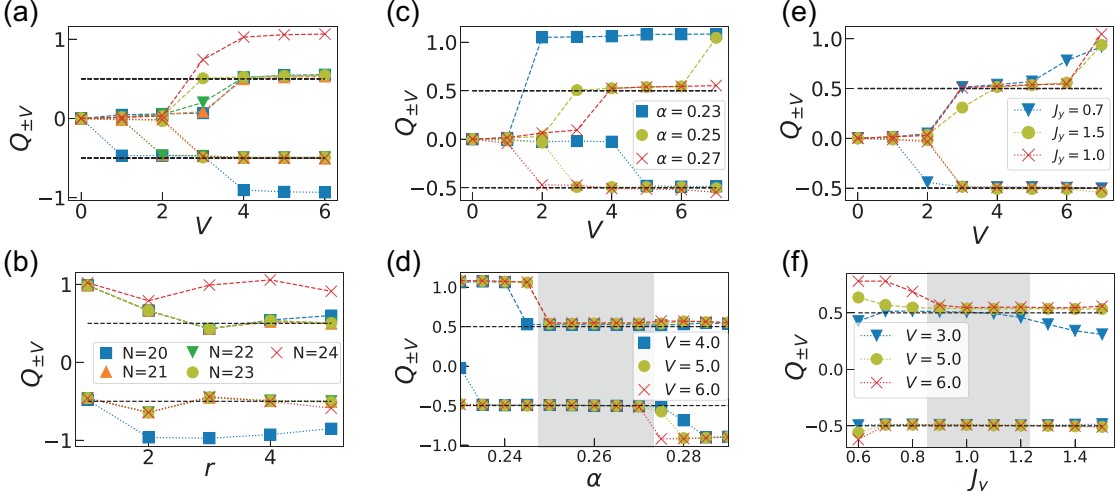

Figure 5: (a) Charges integrated at radius $r = 4$ as a function of pinning strength $V$ for different particle numbers $N$. (b) Charges as a function of $r$ at $V = 5$ for different $N$. We use the same legend in (a) and (b) and $\alpha = 1/4, J_y = 1$. (c) Charges as a function of $V$ for different $\alpha$ and (d) charges as a function of $\alpha$ for different $V$ with $N = 23, J_y = 1$. (e) Charges as a function of $V$ for different $J_y$ and (f) charges as a function of $J_y$ for different $V$ with $N = 23, \alpha = 1/4$. In all cases we have simultaneously applied both negative and positive four-site pinning potentials in a system of size $21 \times 10$.

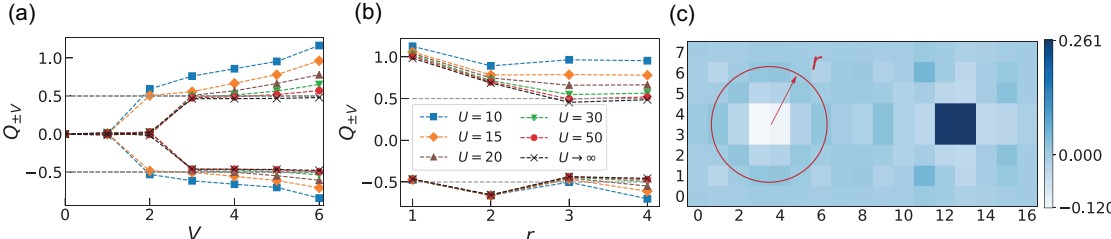

Figure 6: (a) Change of charge induced by a pair of $2 \times 2$ pinning potentials of opposite strength $\pm V$, measured in two circular regions of radius $r$ centered around the pinning potentials. (b) Same as (a), but for fixed $V = 5$ and different radii $r$. We use the same legend for interaction $U$ in (a) and (b). Other parameters are $L_x \times L_y = 17 \times 8$, $\alpha = 1/4$, $J_y = 1$, and $N = (L_x - 1)(L_y - 1)\alpha\nu = 14$. (c) A typical charge distributions at $V = 5$ for $U = 20$. The particle number per site was truncated to a maximum value of 4 (allowing for larger occupations did not change the results).

# 6 Fractional charge pumping

As another hallmark of quantum Hall states, quantized charge pumping can be induced by quanta of adiabatic flux insertion [100,101]. The realization of this famous Laughlin gedanken-experiment in 2D optical lattices has been proposed [102] and its application in interacting systems of small size has also been addressed [46]. In this section, we confirm that such a local-flux insertion can also be exploited to create and to manipulate fractional excitations in 2D FCIs.

After applying additional phases $\delta\phi$ on the links between a target plaquette and the system

boundary, as sketched in Fig. 7(a), only the flux of the target plaquette is modified to be $\phi + \delta\phi$. After linearly ramping $\delta\phi$ from 0 to $2\pi$ within time $\tau$, as expected, a fractional charge of $1/2$ is populated from the bulk to the edge of the system with $\nu = 1/2$ particle per flux quantum [Figs. 7(c,d)]. After the flux insertion, the created edge excitation follows a chiral motion, which is robust against the corner defects [Figs. 7(e-g)]. Note that all these signatures survive

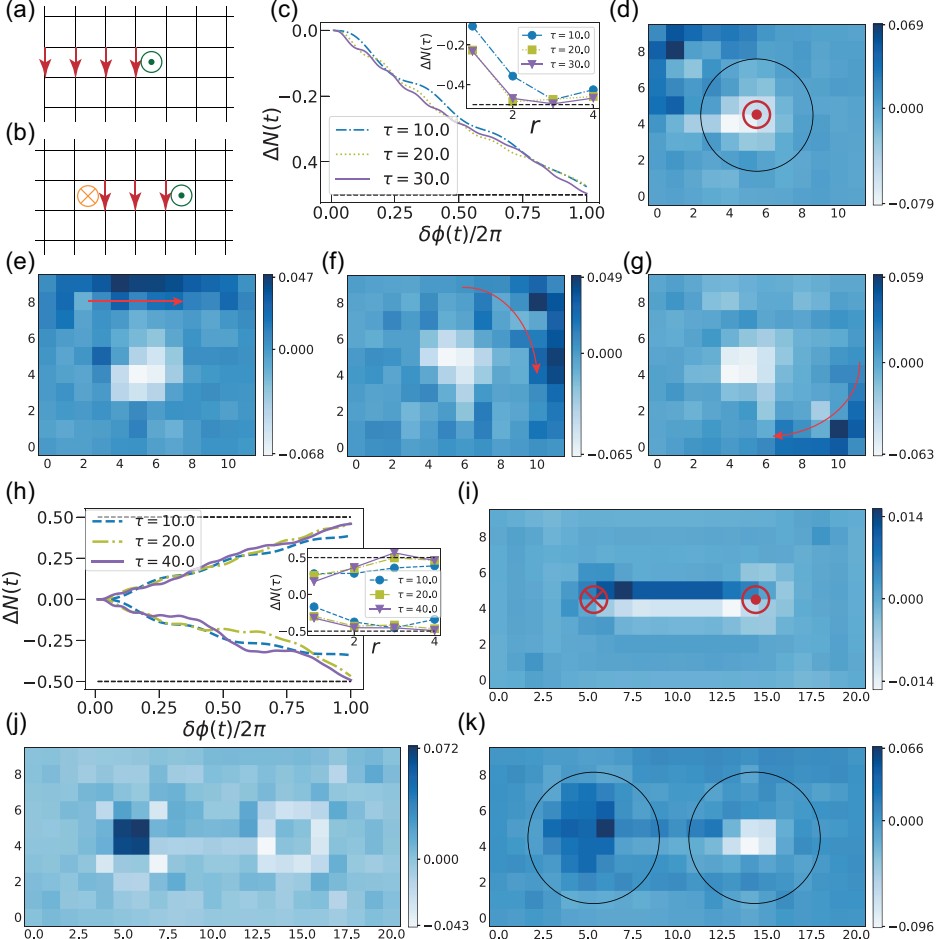

Figure 7: Sketch of modifying the flux in one single plaquette (a) and two plaquettes (b) in the bulk. The arrows denote additional tunnelling phases $\delta\phi$, and the fluxes of the plaquettes labelled by $\odot, \otimes$ are modified to be $\phi \pm \delta\phi$ respectively. (c) The particle number difference ('charge') $\Delta N(t) = N(t) - N(t = 0)$ integrated in the disc $r = 3$ centred at the modified plaquette as a function of $\delta\phi(t)$. The inset shows $\Delta N(\tau)$ as a function of integrated radius $r$. (d) Snapshots of the respective changes in spatial densities at the end of ramp with $t = \tau = 30$ (d), and after the flux insertion at (e) $t = 46$, (f) $t = 71$, (g) $t = 90$. The red arrows are used to indicate the chiral motion. Simulations of (c-g) are performed for a system of size $12 \times 10$ with $N = 12, \alpha = 1/4, J_y = 1$ and $\nu = N/N_\phi \simeq 1/2$. Tiny pinning potentials of strength $V = 1/4$ have been distributed over the four sites of the modified plaquette to prevent the bulk excitations from dispersing. (h) Particle number difference $\Delta N(t)$ integrated in the disc with $r = 4$ centred at the modified plaquettes as a function of $\delta\phi(t)$. The dependence of charges $\Delta N(\tau)$ as a function of $r$ are shown in the inset. Snapshots of the respective density changes within the ramp at (i) $t = 4$, (j) $t = 20$, (k) $t = \tau = 40$. Simulations of (h-k) are performed for a system of size $21 \times 10$ with $N = 23, \alpha = 1/4, J_y = 1$ and $\nu = N/N_\phi \simeq 1/2$.

even for a system of $12 \times 10$ with 12 particles, which is a promising setup within the reach of present-day's experiments [25]. Last but not least, by modifying the fluxes of two plaquettes to be $\phi \pm \delta\phi$, as depicted in Fig. 7(b) and shown in Figs. 7(h-k), QP and QH excitations with charges $\pm 1/2$ can be created in the bulk. Interestingly, we observe fluctuating densities (like a ring structure) in the vicinity of modified plaquettes during the ramp [Fig. 7(j)], which is different from the quantized charge pumping in integer Chern insulators [102].

# 7 Conclusion and outlook

We have numerically investigated the fate of FCI states as the ground state of the hard-core bosonic Harper-Hofstadter model in realistic finite system geometries with open boundary conditions, which can be realized in quantum-gas microscopes. Already for small system sizes starting from linear extents of about 6-8 lattice sites, we find robust measurable signatures that are consistent with the expected behavior of a Laughlin-like FCI state at filling $\nu = 1/2$. In particular, we find chiral edge transport, an incompressible bulk at the expected filling, as well as fractionally charged QP and QH excitations that can be created either by pinning potentials or via local flux insertion. Thanks to the fact that the edges of the system serve as particle reservoirs, these features are rather robust against modifications of both the plaquette flux and the total particle number. Also finite tunneling anisotropies are not detrimental. Our results provide a guide for future experiments with interacting atoms in optical lattices with artificial magnetic flux.

# Acknowledgements

We thank Nathan Goldman, Julian Leonard, Xikun Li, Anne E. B. Nielsen, Frank Pollmann, Hong-Hao Tu, F. Nur Ünal and Wei Wang for fruitful discussions.

**Funding information**  B. W. and A. E. acknowledge support from the Deutsche Forschungs-gemeinschaft (DFG) via the Research Unit FOR 2414 under Project No. 277974659. X. Y. D. is supported by the European Research Council under the grant ERQUAF (715861).

# A Counter-propagating currents and imbalance measurement

To further confirm the existence of robust counter-propagating probability currents as shown in Fig. 1 in the main text, here we provide more data obtained from simulating a larger system of size $21 \times 10$. In Figs. 8(a-d), we plot the distributions of both density and currents of the ground states for different particle numbers $N$. The densities of the middle row $n = 4$ and the vertical currents on the middle links connected by $n = 4$ and $n = 5$ are plotted in Figs. 8(e) and (f), respectively, which represent more pronounced signatures of FCI ground states in the larger system considered here. While the currents in row $n = 1$ change their directions with increasing $N$, the direction of currents in row $n = 2$ remains fixed, and they are always in opposite direction with the currents in the outermost edges. From Fig. 8(g), one can see the amplitude of the counter-propagating currents are almost independent of $N$, which indicate their robust existence. By initially trapping extra one [Fig. 8(h)] or three [Fig. 8(i)] particles in the centre of given row, the density imbalances formed after the quench of on-site trapping present clear negative imbalance on the row $n = 2$ which could be readily measured in experiments.

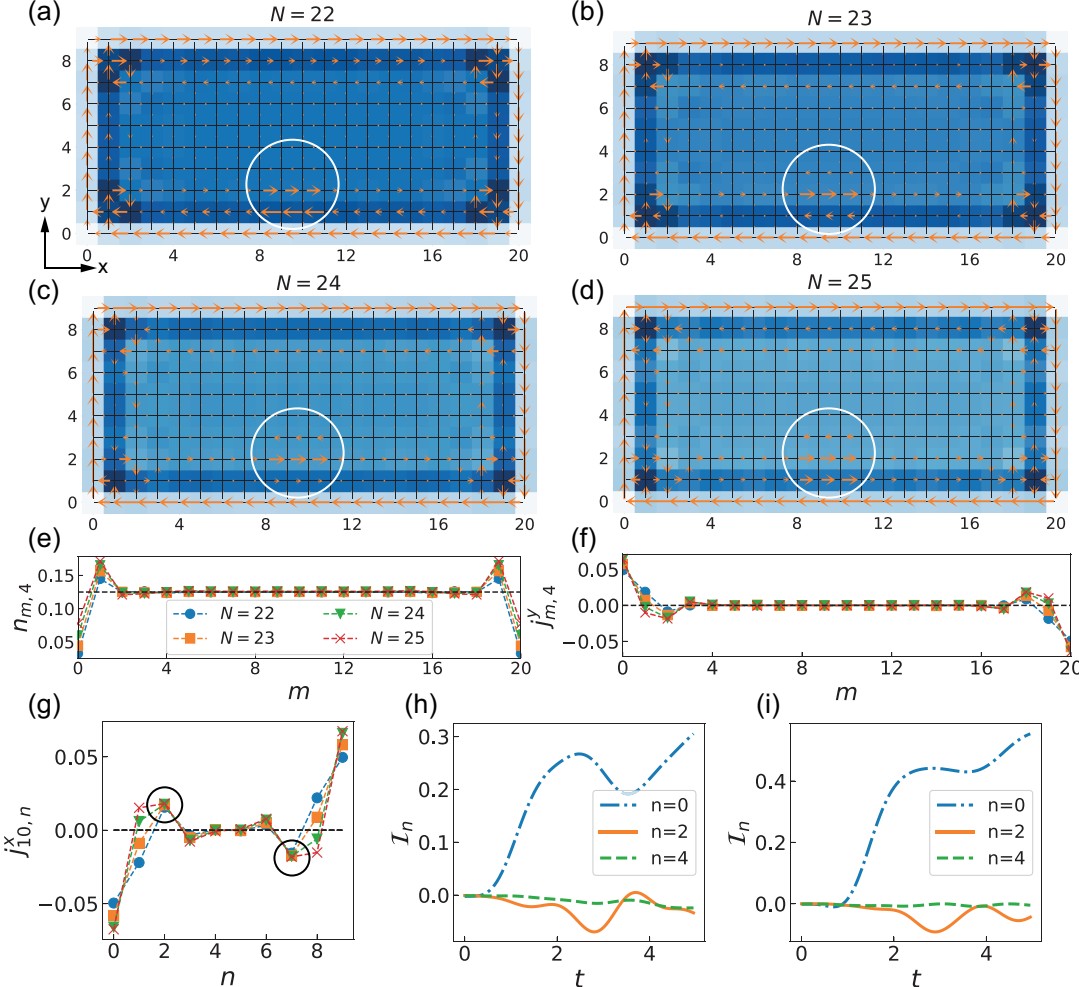

Figure 8: (a-d) Spatial density and current distributions of the ground state of hardcore bosons in a lattice of size $21 \times 10$ with $N = 22, 23, 24, 25$ respectively. The magnitudes of currents in the white circle are zoomed in by a factor of 3 for a clear visualization. (e) Density of the middle row $n = 4$ and (f) vertical currents on the middle links connected by $n = 4$ and $n = 5$. The horizontal dashed lines locates the expected $\rho = 1/8$ and $j = 0$. (g) Horizontal currents on the middle bonds connected by $m = 10$ and $m = 11$. (e-g) share the same legend. The black circles are used to highlight that the currents with opposite chirality are almost independent of $N$. Imbalance in the $n$-th row $\mathcal{I}_n$ as a function of evolution time with (h) one and (i) three extra particles initially trapped in the centre of given rows. Consistent negative $\mathcal{I}_2$ appear after roughly three tunneling time. Other parameters are $\alpha = 1/4$ and $J_y = 1$.

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
