# Peer review of "Measurable signatures of bosonic fractional Chern insulator states and their fractional excitations in a quantum-gas microscope"

_SciPost Physics, doi:SciPost Phys. 12, 095 (2022)_

## Round 1 · Referee Report · Anonymous (Referee 1) · 2021-12-9

Strengths

1- The manuscript is well written and easily readable 2- This is an active research field which attracts a lot of attention 3- DMRG calculations provide an advantage over previous exact diagonalization studies 4- The results could be of use for future experiments

Weaknesses

1- There is already an abundance of papers on bosonic FCI states in the Harper-Hofstadter model 2- Finite interaction strengths are only briefly considered 3- Some minor issues need to be addressed

Report

In this manuscript, the Authors study the Harper-Hofstadter model in the strongly interacting regime. This system is expected to host bosonic fractional Chern insulator (FCI) states. They compute several experimentally measurable quantities which could serve as signatures of the FCI state, such as incompressible bulk density and vanishing particle currents. They also apply local pinning potentials and investigate the formation of anyonic excitations with fractional charge. Moreover, they determine the necessary conditions for observing the signatures of the FCI state and the parameter regimes where this state is robust by varying the lattice size, particle number, magnetic flux per plaquette, tunneling anisotropy and other parameters.

Although the idea of realizing bosonic FCI states in optical lattices with artificial gauge fields and strong interaction is not new and has already been extensively studied, this manuscript provides some new insights. The Authors perform numerical simulations based on DMRG, which allows them to study larger system sizes and gives them an advantage over some previous works which were based on exact diagonalization. Significant progress in engineering artificial gauge fields in optical lattices has been made in recent years, as well as in development of quantum gas microscopy, which gives hope that bosonic FCI states could be indeed experimentally realized and probed. This work could provide guidance for such future experiments.

To summarize, I recommend the publication of this paper in SciPost Physics. The manuscript is well written and will surely be of interest to other researchers in this field. All claims are supported by clear numerical evidence. However, I would like the Authors to address several minor issues before publication.

Requested changes

1- Introduction, second paragraph: "Various realistic protocols were proposed for both the detections of the FCT states..." Do you mean FCI instead of FCT? If not, the acronym FCT needs to be defined. 2- Discussion of Fig.1(b): It is said that the FCI state breaks down when $N\geq17$, but it is not shown what happens when $N<14$. What is the lower limit for the particle number needed to obtain the FCI state for this system size? 3- Page 5: "the current at the third row away from the boundary is opposite to the current at the boundary." Is it the third or the second row? Fig.1(c) suggests that this happens for $n=2$. If $n=0$ is the boundary, then $n=1$ is the first row away from the boundary and $n=2$ is the second row. 4- Fig.2(a): There is no colour scale. Is it the same as in Fig.1(a)? 5- Fig.3: What is the reason for using two different shapes of pinning potentials (2x2 and 3x1)? Can you comment on the differences between these two cases? Figs.3(b) and (e) suggest that the 3x1 potential is less sensitive to different values of $r$ and $N$ (at least for $r\geq3$ and $N\geq6$), most likely due to its smaller extent in the y-direction, but the 2x2 potential is used in later figures. Additionally, Figs.3(c) and (f) are referred to in the main text before all other panels in this figure. Maybe it would make sense to move these two panels to the left and change their labels to (a) and (d)? 6- Inset of Fig.4(a): What is the difference between the squares and the circles? Is the value of $V$ lower for the circles (it seems to be below the threshold)? 7- Fig.6(c): Are the y-labels supposed to be $\Delta N(t)$ in the main figure and $\Delta N(\tau)$ (charge after the ramp) in the inset? If not, to what time $t$ do the data points in the inset correspond? Same comment for Fig.6(h). Also, Figs.6(h-k) are not referred to at all in the main text. 8- Figs.7(e) and (f) are not mentioned at all in the text of Appendix A. Do we learn anything new from this figures that was not clear from Fig.1(b)? Maybe it could be said that the FCI signatures are more pronounced (flatter density and current distributions) for this system size than what was shown in Fig.1(b)? 9- Suggestion: It would be interesting to see a more detailed study of the effects of finite interactions. The main text focuses on hard-core bosons and the current Appendix B is very short. This is an important point since the Authors wish to simulate realistic experimental conditions. If this system is experimentally realized, the interaction strength in the experiment will of course be finite, and it might be preferable not to increase the interaction strength more than is necessary as it could lead to unwanted heating. It would be useful to show the effects of interactions on some other quantities, for example Fig.1(b) with different interaction strengths instead of different particle numbers.

There are also several small typos: 1- In the References, many titles are not properly capitalized and should be corrected. For example, “rydberg atoms” → “Rydberg atoms”, “haldane model” → “Haldane model”, “bose-einstein condensates” → “Bose-Einstein condensates”, and many others. 2- Caption of Fig.1(c): “black circle are used” → “black circle is used” 3- Page 5: “typical currents ... is plotted” → “typical currents ... are plotted” 4- Two different notations ($\mathcal{I}_n$ and $I_n$) are used for the density imbalance in the paragraph around Eq.(4). Please make the notation consistent. 5- Caption of Fig.3(c): “red circle are used” → “red circles are used”

  • validity: high
  • significance: high
  • originality: good
  • clarity: top
  • formatting: good
  • grammar: excellent

Author:  Botao Wang  on 2021-12-23  [id 2048]

(in reply to Report 1 on 2021-12-09)

Thank you very much for the constructive comments and suggestions. Please kindly find the attached pdf file for our reply.

Attachment:

Reply_Report_1.pdf

---

## Round 1 · Referee Report · Tobias Grass (Referee 2) · 2021-12-16

Strengths

1) timely subject: The addressed questions are very relevant in view of existing experimental efforts in the field 2) applied theoretical methods (DMRG) is solid and well-suited to explore the asked questions 3) convincing results 4) clear presentation of the results

Weaknesses

1) Results, at least qualitatively, are rather expected.

Report

The paper studies signatures of fractional Chern insulators (FCI) which could be detected using optical lattice microscopy. There are continueing experimental efforts to realize such a phase with cold atoms in optical lattices, but it remains challenging to reach larger system sizes (i.e. of a few rather than two atoms). The present manuscript focuses on systems of a scale (10-20 atoms) which seems to be realistic in the near future, but for which it is not a priori clear how well the system would actually exhibit the expected bulk behavior. At the same time, such a system size is already challenging for classical simulations, which the authors manage by using DMRG. The obtained results demonstrate very convincingly that, even for those modest sizes, the expected bulk behavior (flat density, fractionally charged excitations, etc) could be observed. These are relevant and solid findings which I recommend to publish on SciPost.

Requested changes

1) The definition of "filling" (nu) is only given in Section 3, but the term is already used in abstract and introduction. Since there are two different types of fillings (nu and rho), there might be some confusion. In fact, a precise definition of nu is never given (only: nu APPROX N/N_Phi). Of course, there are finite-size effects, but in the thermodynamic limit nu should be sharply defined.

2) Section 2 mentions that results are for hard-core bosons, but only hold for finite U. Here the authors should refer to their supplemental material (Part B) which discusses finite U. The statement in Sec. 2 reads a bit as if all results would hold for essentially arbitrary values of U -- which of course is not the case.

3) What motivates the choice of alpha=1/4? The authors should explain that this is not a random choice.

4) In Fig. 1d, the value I_n is claimed to be consistent with the current at "short times" after the quench. What does the authors mean by "short times"? The n=2 curve oscillates between something negative (consistent) and zero (not consistent). So maybe the statement that the time-averaged I_n is consistent with the sign of the current would be more accurate? Apart from the sign, is it possible to extract the quantitative value of the currents? Can the quantization of currents be measured in that way?

5) In Part B, the authors say that "converging" behavior has been observed by choosing larger simulation parameters. Does it mean that the shown results have not yet converged? Why don't they show the results for larger xi?

6) Typos: second paragraph introductions FCT -> FCI page 5: currents ... "are" plotted versus the y-coordinate

7) The authors may consider adding these references: - Eliot Kapit, Paul Ginsparg, and Erich Mueller, Phys. Rev. Lett. 108, 066802 (2012) - Tobias Graß, Bruno Juliá-Díaz, and Maciej Lewenstein, Phys. Rev. A 89, 013623 (2014)

  • validity: top
  • significance: high
  • originality: good
  • clarity: top
  • formatting: excellent
  • grammar: excellent

Author:  Botao Wang  on 2021-12-23  [id 2049]

(in reply to Report 2 by Tobias Grass on 2021-12-16)
Category:
answer to question
correction

Thank you very much for the constructive comments and suggestions. Please kindly find the attached pdf file for our reply.

Attachment:

Reply_Report_2.pdf

---

## Round 1 · Referee Report · Anonymous (Referee 3) · 2021-12-19

Strengths

1- Timely study 2- Reliable state-of-the-art numerics 3- Thorough study of finite-size effects, complementary with other works

Weaknesses

1- Several works have addressed the detection of FCIs in optical lattices through similar methods, as referenced in the manuscript

Report

This manuscript explores the detection of fractional Chern insulator (FCI) states in optical lattices through local density and current measurements, as provided by quantum gas microscopes. The model used for this study is the Bose-Hubbard Harper-Hofstadter model on the square lattice, focusing on a small magnetic flux window around pi/2 per plaquette. The Harper-Hofstadter model has been realized experimentally by several groups, and is thus a very common model for theoretical studies aiming to guide experiments towards the realization of FCIs. The authors focus mostly on hard-core bosons, but also include an estimation of finite U effects in the appendices. The authors have considered a system with edges and infinite wall confinement potential, which is consistent with some existing experiments. Their results are based on numerical calculations with DMRG. The authors have explored the local charge and current densities in the system, in the ground state as well as upon addition of one (or two oppositely) charged impurity potentials. Integrating the charge in a small region around the impurity yields a charge defect +/-e/2, as expected for Laughlin quasiparticles. Moreover, the authors have implemented a local (one plaquette) flux insertion numerically, and observed the expected quantized fractional charge pumping.

As pointed out by the authors themselves, these methods are not conceptually new, but have been proposed and even numerically explored before. The big strength of this manuscript is to thoroughly explore the finite-size effects (and robustness with respect to microscopic parameters) associated with each method. For every explored phenomenon, the authors provide numerical data for systems as large as can be simulated by state-of-the-art DMRG (which the data convinces us is large enough to enter a universal regime), as well as smaller systems, down to the small sizes where these universal phenomena break down. While the large size simulations are necessary to convince us of the universality of the results, the small size data is equally useful in view of experimental implementation, and the data in-between also provides useful information about the emergence of FCIs. In that sense, the results shown here nicely complement and confirm former studies (for example, the smallest sizes needed to observe universal FCI phenomena are in agreement with previous works). In conclusion, I recommend publication of this manuscript in Scipost Physics.

Requested changes

1- When discussing the addition of a QH-QP pair, the authors write 'The choice of such a pair of defects is motivated by the desire to keep the average filling away from the defects constant.' But later, they perform the same numerical experiment with just one hole. Can you please comment further on this? Clearly, adding just one QH (or QP) is favorable from the point of view of finite-size effects since the system only needs to accomodate one QH/QP instead of a pair (which additionally has to be sufficiently separated). Since the result seems to be the same (fractionally quantized charge), is there any real advantage to adding a QH-QP pair (vs just one QH or QP)? And in that case, would it be a good idea to place one defect delocalized around the edge and one defect in the center to maximize distance between the two defects for a given system size?

2- fig. 3: QH and QP (upper and lower pannels of Fig b and e) should have same scale and y tics, so that it is easier to compare them to one another.

3- Inset of fig. 4a what are the two types of orange dots?

4- fig. 4 b 'The currents within the circle are zoomed in for clear visualization.' Please specify by how much the currents are enhanced (same thing for other zooms in other figures)

5- p. 8, please fix the typo ' to estimate the the spatial extent of QHs' (two times 'the')

  • validity: top
  • significance: high
  • originality: good
  • clarity: high
  • formatting: excellent
  • grammar: excellent

Author:  Botao Wang  on 2021-12-23  [id 2050]

(in reply to Report 3 on 2021-12-19)
Category:
answer to question
correction

Thank you very much for the constructive comments and suggestions. Please kindly find the attached pdf file for our reply.

Attachment:

Reply_Report_3.pdf

---

## Round 2 · Referee Report · Anonymous (Referee 1) · 2021-12-30

Report

I have reviewed the revised version and I am satisfied by the Authors’ response to my questions and comments. However, there is still one typo in Appendix A. The sentence “The densities of the middle row $n = 4$ and the vertical currents on the middle links connected by $n = 4$ and $n = 5$ are plotted in Figs. 8(c) and (d)…” should instead refer to Figs. 8(e) and (f). Other than that, the manuscript is now ready for publication in SciPost Physics.
  • validity: high
  • significance: high
  • originality: good
  • clarity: top
  • formatting: excellent
  • grammar: excellent

Author:  Botao Wang  on 2022-01-07  [id 2077]

(in reply to Report 1 on 2021-12-30)

Thank you for close reading and the useful comment. The typo has now been corrected in the resubmitted manuscript.

---

## Round 2 · Author Response

Dear Editor,

We would like to resubmit our revised paper to SciPost Physics.

Our manuscript has been reviewed by three referees who all recommended the publication after minor modifications. We thank the referees for close reading and also for their constructive comments and suggestions. All the points have been addressed in our replies to the referees, and the manuscript has been modified accordingly. We hope our revised paper would be suitable for publication.

Sincerely,
Botao Wang, Xiaoyu Dong, and André Eckardt

---

## Round 3 · Author Response

Dear Editor,

Thank the referee for close reading and the final comment. It has been implemented in the resubmitted manuscript. We hope our revised paper is now ready for publication.

Sincerely,
Botao Wang, Xiaoyu Dong, and André Eckardt

---

## Editorial Decision

published